# COUNTERFACTUAL GENERATION UNDER CONFOUNDING

## ABSTRACT

A machine learning model, under the influence of observed or unobserved confounders in the training data, can learn spurious correlations and fail to generalize when deployed. For image classifiers, augmenting a training dataset using counterfactual examples has been empirically shown to break spurious correlations. However, the counterfactual generation task itself becomes more difficult as the level of confounding increases. Existing methods for counterfactual generation under confounding consider a fixed set of interventions (e.g., texture, rotation) and are not flexible enough to capture diverse data-generating processes. Given a causal generative process, we formally characterize the adverse effects of confounding on any downstream tasks and show that the correlation between generative factors (attributes) can be used to quantitatively measure confounding between generative factors. To minimize such correlation, we propose a counterfactual generation method that learns to modify the value of any attribute in an image and generate new images given a set of observed attributes, even when the dataset is highly confounded. These counterfactual images are then used to regularize the downstream classifier such that the learned representations are the same across various generative factors conditioned on the class label. Our method is computationally efficient, simple to implement, and works well for any number of generative factors and confounding variables. Our experimental results on both synthetic (MNIST variants) and real-world (CelebA) datasets show the usefulness of our approach.

## 1 INTRODUCTION

A confounder is a variable that causally influences two or more variables that are not necessarily directly causally dependent (Pearl, 2001). Often, the presence of confounders in a data-generating process is the reason for spurious correlations among variables in the observational data. The bias caused by such confounders is inevitable in observational data, making it challenging to identify invariant features representative of a target variable (Rothenhäusler et al., 2021; Meinshausen & Bühlmann, 2015; Wang et al., 2022). For example, the *demographic area* an individual resides in often confounds the *race* and perhaps the level of *education* that individual receives. Using such observational data, if the goal is to predict an individual's *salary*, a machine learning model may exploit the spurious correlation between *education* and *race* even though those two variables should ideally be treated as independent variables. Removing the effects of confounding in trained machine learning models has shown to be helpful in various applications such as zero or few-shot learning, disentanglement, domain generalization, counterfactual generation, algorithmic fairness, healthcare, etc. (Suter et al., 2019; Kilbertus et al., 2020; Atzmon et al., 2020; Zhao et al., 2020; Yue et al., 2021; Sauer & Geiger, 2021; Goel et al., 2021; Dash et al., 2022; Reddy et al., 2022; Dinga et al., 2020).

In observational data, confounding may be observed or unobserved and can pose various challenges in learning models depending on the task. For example, disentangling spuriously correlated features using generative modeling when there are confounders is challenging (Sauer & Geiger, 2021; Reddy et al., 2022; Funke et al., 2022). As stated earlier, a classifier may rely on non-causal features to make predictions in the presence of confounders (Schölkopf et al., 2021). Recent years have seen a few efforts to handle the spurious correlations caused by confounding effects in observational data (Träuble et al., 2021; Sauer & Geiger, 2021; Goel et al., 2021; Reddy et al., 2022). However, these methods either make strong assumptions on the underlying causal generative process or require strong supervision. In this paper, we study the adversarial effect of confounding in observational data

on a classifier's performance and propose a mechanism to marginalize such effects when performing data augmentation using counterfactual data. Counterfactual data generation provides a mechanism to address such issues arising from confounding and building robust learning models without the additional task of building complex generative models.

The causal generative processes considered throughout this paper are shown in Figure 1(a). We assume that a set of generative factors (attributes) $Z_1, Z_2, \ldots, Z_n$ (e.g., *background, shape, texture*) and a label $Y$ (e.g., *cow*) *cause* a real-world observation $X$ (e.g., an image of a cow in a particular background) through an unknown causal mechanism $g$ (Peters et al., 2017b). To study the effects of confounding, we consider $Y, Z_1, Z_2, \ldots, Z_n$ to be confounded by a set of confounding variables $C_1, \ldots, C_m$ (e.g., certain breeds of *cows* appear only in certain *shapes* or *colors* and appear only in certain *countries*). Such causal generative processes have been considered earlier for other kinds of tasks such as disentanglement Suter et al. (2019); Von Kügelgen et al. (2021); Reddy et al. (2022). The presence of confounding variables results in spurious correlations among generative factors in the observed data, whose effect we aim to remove using counterfactual data augmentation.

A related recent effort by (Sauer & Geiger, 2021) proposes Counterfactual Generative Networks (CGN) to address this problem using a data augmentation approach. This work assumes each image to be composed of three Independent Causal Mechanisms (ICMs) (Peters et al., 2017a) responsible for three fixed factors of variations: *shape, texture*, and *background* (as represented by $Z_1, Z_2$, and $Z_3$ in Figure 1(b)). This work then trains a generative model that learns three ICMs for *shape, texture*, and *background* separately, and combines them in a deterministic fashion to generate observations. Once the ICMs are learned, sampling images by making interventions to these mecha-

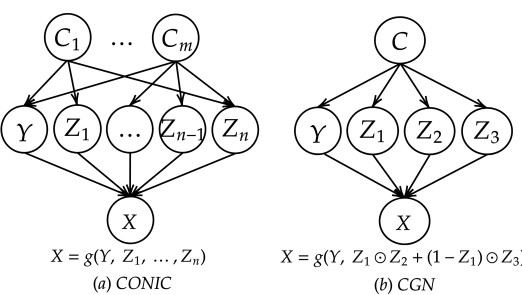

$X = g(Y, Z_1, \ldots, Z_n)$     $X = g(Y, Z_1 \odot Z_2 + (1 - Z_1) \odot Z_3)$
      *(a) CONIC*                  *(b) CGN*

Figure 1: *(a)* causal data generating process considered in this paper (CONIC = Ours); *(b)* causal data generating process considered in CGN (Sauer & Geiger, 2021).

nisms give counterfactual data that can be used along with training data to improve classification results. However, fixing the architecture to specific number and types of mechanisms (*shape, texture, background*) is not generalizable, and may not directly be applicable to settings where the number of underlying generative factors is unknown. It is also computationally expensive to train different generative models for each aspect of an image such as *texture, shape* or *background*.

In this work, we begin with quantifying confounding in observational data that is generated by an underlying causal graph (more general than considered by CGN) of the form shown in Figure 1(a). We then provide a counterfactual data augmentation methodology called CONIC (COunterfactual geNeratIon under Confounding). We hypothesize that the counterfactual images generated using the proposed CONIC method provide a mechanism to marginalize the causal mechanisms responsible for spurious correlations (i.e., causal arrows from $C_i$ to $Z_j$ for some $i, j$). We take a generative modeling approach and propose a neural network architecture based on conditional CycleGAN (Zhu et al., 2017) to generate counterfactual images. The proposed architecture improves CycleGAN's ability to generate quality counterfactual images under confounded data by adding additional contrastive losses to distinguish between fixed and modified features, while learning the cross domain translations. To demonstrate the usefulness of such counterfactual images, we consider classification as a downstream task and study the performance of various models on unconfounded test set. Our key contributions include:

- We formally quantify confounding in causal generative processes of the form in Fig 1(a), and study the relationship between correlation and confounding between any pair of generative factors.

- We present a counterfactual data augmentation methodology to generate counterfactual instances of observed data, that can work even under highly confounded data ($\sim 95\%$ confounding) and provides a mechanism to marginalize the causal mechanisms responsible for confounding.

- We modify conditional CycleGAN to improve the quality of generated counterfactuals. Our method is computationally efficient and easy to implement.

- Following previous work, we perform extensive experiments on well-known benchmarks – three MNIST variants and CelebA datasets – to showcase the usefulness of our proposed methodology in improving the accuracy of a downstream classifier.

## 2 RELATED WORK

**Counterfactual Inference:** (Pearl, 2009), in his seminal text on causality, provided a three-step procedure for generation of a counterfactual data instance, given an observed instance: (i) *Abduction:* abduct/recover the values of exogenous noise variables; (ii) *Action:* perform the required intervention; and (iii) *Prediction:* generate the counterfactual instance. One however needs access to the underlying structural causal model (SCM) to perform the above steps for counterfactual generation. Since real-world data do not come with an underlying SCM, many recent efforts have focused on modeling the underlying causal mechanisms generating data under various assumptions. These methods then perform the required intervention on specific variables in the learned model to generate counterfactual instances that can be used for various downstream tasks such as classification, fairness, explanations etc. (Kusner et al., 2017; Joo & Kärkkäinen, 2020; Denton et al., 2019; Zmigrod et al., 2019; Pitis et al., 2020; Yoon et al., 2018; Bica et al., 2020; Pawlowski et al., 2020).

**Generating Counterfactuals by Learning ICMs:** In a more recent effort, assuming any real-world image is generated with three independent causal mechanisms for *shape, texture, background*, and a *composition* mechanism of the first three, (Sauer & Geiger, 2021) developed Counterfactual Generative Networks (CGN) that generate counterfactual images of a given image. CGN trains three Generative Adversarial Networks (GANs) (Goodfellow et al., 2014b) to learn *shape, texture, background* mechanisms and combine these three mechanisms using a composition mechanism $g$ as $g(shape, texture, background) = shape \odot texture + (1 - shape) \odot background$ where $\odot$ is the Hadamard product. Each of these independent mechanisms is given an input of noise vector $u$ and a label $y$ specific to that independent mechanism while training. Once the independent mechanisms are trained, counterfactual images are generated by sampling a label and a noise vector corresponding to each mechanism and then feeding the input to CGN. Finally, a classifier is trained with both original and counterfactual images to achieve better test time accuracy, showing the usefulness of CGN. However, such deterministic nature of the architecture is not generalizable to the case where the number of underlying generative factors are unknown and it is computationally infeasible to train generative models for specific aspect of an image such as texture/background.

**Disentanglement and Data Augmentation:** The spurious correlations among generative factors have been considered in disentanglement (Funke et al., 2022; von Kügelgen et al., 2021). The general idea in these efforts is to separate the causal predictive features from non-causal/spurious predictive features to predict an outcome. Our goal is different from disentanglement, and we focus on the performance of a downstream classifier instead of separating the sources of generative factors. Traditional data augmentation methods such as rotation, scaling, corruption, etc. (Hendrycks et al., 2020; Devries & Taylor, 2017; Zhang et al., 2018; Yun et al., 2019) do not consider the causal generative process and hence they can not remove the confounding in the images via data augmentation (e.g., color and shape of an object can not be separated using simple augmentations). We hence focus on counterfactual data augmentations that is focused on marginalizing the confounding effect caused by confounders.

A similar effort to our paper is by (Goel et al., 2021) who use CycleGAN to generate counterfactual data points. However, they focus on the performance of a subgroup (a subset of data with specific properties) which is different from our goal of controlling confounding in the entire dataset. Another recent work by (Wang et al., 2022) considers spurious correlations among generative factors and uses CycleGAN to generate counterfactual images. Compared to these efforts, rather than using CycleGAN directly, we propose a CycleGAN-based architecture that is optimized for *controlled* generation using contrastive losses.

**Applications of Counterfactuals:** Augmenting the training data with appropriate counterfactual data has shown to be helpful in many applications ranging from vision to natural language tasks (Joo & Kärkkäinen, 2020; Lample et al., 2017; Kusner et al., 2017; Kaushik et al., 2019; Dash et al., 2022). (Joo & Kärkkäinen, 2020) identified existing biases in computer vision APIs deployed in the real world by Amazon, Google, IBM, and Clarifai by looking at the differences made by those APIs on counterfactual images that differ by protected/sensitive attributes (e.g., race and gender). Using locally independent causal mechanisms, (Pitis et al., 2020) augmented training data with counterfactual data points in a model-free reinforcement learning setting. Here, the idea is to use any two factual trajectories of an episode and combine the two trajectories at a particular point in time to generate the counterfactual data point, which will then be added to the replay buffer. Independently factored samples are essential to get plausible and realistic counterfactual instances.

## 3 INFORMATION THEORETIC MEASURE OF CONFOUNDING

**Background and Problem Formulation:** Let $\{Z_1, Z_2, \ldots, Z_n\}$ be a set of random variables denoting the generative factors of an observed data point $X$, and $Y$ be the label of the observation $X$. Each generative factor $Z_i$ (e.g., *color*) can take on a value form a discrete set of values $\{z_i^1, \ldots, z_i^d\}$ (e.g., *red, green* etc.). Let the set $S = \{Y, Z_1, \ldots, Z_n\}$ generates $N$ real-world observations $\{X_i\}_{i=1}^N$ through an unknown causal mechanism $g$. Each $X_i$ can be thought of as an observation generated using the causal mechanism $g$ with certain intervention on the variables in the set $S$. Variables in $S$ may potentially be confounded by a set of confounders $C = \{C_1, \ldots, C_m\}$ that denote real-world confounding such as selection bias. Let $\mathcal{D}$ be the dataset of real-world observations along with corresponding values taken by $\{Y, Z_1, \ldots, Z_n\}$. Causal graph in Figure 1(a) shows the general form of this setting. From a causal effect perspective, each variable in $S$ has a direct causal influence on the observation $X$ (e.g., the causal edge $Z_i \to X$) and also has non-causal influence on $X$ via the confounding variables $C_1, \ldots, C_m$ (e.g., $Z_i \leftarrow C_j \to Z_k \to X$ for some $C_j$ and $Z_k$). These paths via the confounding variables, in which there is an incoming arrow to the variables in $S$, are also referred to as *backdoor paths* (Pearl, 2001). Due to the presence of backdoor paths, we may observe spurious correlations among the variables in $S$ in the observational data $\mathcal{D}$.

In any downstream application where $\mathcal{D}$ is used to train a model (e.g., classification, disentanglement etc.), it is desirable to minimize or remove the effect of confounding variable to ensure that a model is not exploiting the spurious correlations in the data to arrive at a decision. In this paper, we present a method to remove the effect of such confounding variables using counterfactual data augmentation. We start by studying the relationship between the amount of confounding and the correlation between any pair of generative factors in causal processes of the form shown in Figure 1(a).

**Definition 3.1.** *No Confounding (Pearl, 2009). In a causal directed acyclic graph (DAG) $\mathcal{G} = (\mathcal{V}, \mathcal{E})$, where $\mathcal{V}$ denotes the set of variables and $\mathcal{E}$ denotes the set of directed edges denoting the direction of causal influence among the variables in $\mathcal{V}$, an ordered pair $(Z_i, Z_j); Z_i, Z_j \in \mathcal{V}$ is unconfounded if and only if $p(Z_i = z_i | do(Z_j = z_j)) = p(Z_i = z_i | Z_j = z_j), \forall z_i, z_j$. Where $do(Z_i = z_i)$ denotes an intervention to the variable $Z_i$ with the value $z_i$. This definition can also be extended to disjoint sets of random variables.*

Definition 3.1 provides the notion of *no confounding*, however, to quantify the notion of *confounding* between a pair of variables, we consider the following definition that relates the interventional distribution $p(Z_i | do(Z_j))$ and the conditional distribution $p(Z_i | Z_j)$.

**Definition 3.2.** *(Directed Information (Raginsky, 2011; Wieczorek & Roth, 2019)). In a causal directed acyclic graph (DAG) $\mathcal{G} = (\mathcal{V}, \mathcal{E})$, where $\mathcal{V}$ denotes the set of variables and $\mathcal{E}$ denotes the set of directed edges denoting the direction of causal influence among the variables in $\mathcal{V}$, the directed information from a variable $Z_i \in \mathcal{V}$ to another variable $Z_j \in \mathcal{V}$ is denoted by $I(Z_i \to Z_j)$. It is defined as follows.*

$$I(Z_i \to Z_j) := D_{KL}(p(Z_i|Z_j)||p(Z_i|do(Z_j))|p(Z_j)) := \mathbb{E}_{p(Z_i, Z_j)} \log \frac{p(Z_i|Z_j)}{p(Z_i|do(Z_j))} \tag{1}$$

Using Definitions 3.1 and 3.2, it is easy to see that the variables $Z_i$ and $Z_j$ are unconfounded if and only if $I(Z_j \to Z_i) = 0$. Non zero directed information $I(Z_j \to Z_i)$ entails that, $p(Z_i|Z_j) \neq p(Z_i|do(Z_j))$ and hence the presence of confounding (if there is no confounder, $p(Z_i|Z_j)$ should be equal to $p(Z_i|do(Z_j))$). Also, it is important to note that the directed information is not symmetric (i.e., $I(Z_i \to Z_j) \neq I(Z_j \to Z_i)$) (Jiao et al., 2013). We use this fact in defining the measure of confounding below. Since we need to quantify the notion of *confounding* (as opposed to *no confounding*), we use directed information to quantify *confounding* as defined below.

**Definition 3.3.** *(An Information Theoretic Measure of Confounding.) In a causal directed acyclic graph (DAG) $\mathcal{G} = (\mathcal{V}, \mathcal{E})$, where $\mathcal{V}$ denotes the set of variables and $\mathcal{E}$ denotes the set of directed edges denoting the direction of causal influence among the variables in $\mathcal{V}$, the amount of confounding between a pair of variables $Z_i \in \mathcal{V}$ and $Z_j \in \mathcal{V}$ is equal to $I(Z_i \to Z_j) + I(Z_j \to Z_i)$.*

Since directed information is not symmetric, we define the measure of confounding to include the directed information from one variable to the other for a given pair of variables $Z_i, Z_j$. We now relate the quantity $I(Z_i \to Z_j) + I(Z_j \to Z_i)$ with the correlation between generative factors so that it is easy to quantify the amount of confounding in observational data. Before that, we present the following proposition which will be used in the proof of the subsequent proposition.

**Proposition 3.1.** *In causal processes of the form 1(a), the interventional distribution $p(Z_i|do(Z_j))$ is same as the marginal distribution $p(Z_i)$.*

*Proof.* In causal processes of the form 1(a), let $C'$ denote the set of all confounding variables that are part of some backdoor path from $Z_i$ to $Z_j$. That is $C' = \{C|Z_i \leftarrow C \rightarrow Z_j\}$ for some $i, j$. Then we can evaluate the quantity $p(Z_i|do(Z_j))$ as

$$p(Z_i|do(Z_j)) = \sum_{C'} p(Z_i|Z_j, C')p(C') = \sum_{C'} p(Z_i|C')p(C') = \sum_{C'} p(Z_i, C') = p(Z_i)$$

Where the first equality is because of the adjustment formula (Pearl, 2001) and the second equality is because of the fact that $Y$ is a collider in causal graph 1(a) and hence conditioned on $C'$, $Z_i$ is independent of $Z_j$. $\square$

**Proposition 3.2.** *For causal generative processes of the form 1(a), the correlation between a pair of generative factors $(Z_i, Z_j)$ is proportional to the amount of confounding between $Z_i$ and $Z_j$.*

*Proof.* Expanding the quantity $I(Z_i \rightarrow Z_j) + I(Z_j \rightarrow Z_i)$, we get the following,

$$
\begin{aligned}
I(Z_i \rightarrow Z_j) + I(Z_j \rightarrow Z_i) &= \mathbb{E}_{Z_i,Z_j}\left[\log(\frac{p(Z_i|Z_j)}{p(Z_i|do(Z_j))})\right] + \mathbb{E}_{Z_i,Z_j}\left[\log(\frac{p(Z_j|Z_i)}{p(Z_j|do(Z_i))})\right]\\
&= \mathbb{E}_{Z_i,Z_j}\left[\log(\frac{p(Z_i|Z_j)p(Z_j|Z_i)}{p(Z_i|do(Z_j))p(Z_j|do(Z_i))})\right] = \mathbb{E}_{Z_i,Z_j}\left[\log(\frac{p(Z_i|Z_j)p(Z_j|Z_i)}{p(Z_i)p(Z_j)})\right]\\
&= \mathbb{E}_{Z_i,Z_j}\left[\log(\frac{p(Z_i|Z_j)p(Z_j)p(Z_j|Z_i)p(Z_i)}{p(Z_i)p(Z_j)p(Z_i)p(Z_j)})\right] = \mathbb{E}_{Z_i,Z_j}\left[\log(\frac{p(Z_i,Z_j)p(Z_j,Z_i)}{p(Z_i)^2p(Z_j)^2})\right]\\
&= 2 \times \mathbb{E}_{Z_i,Z_j}\left[\log(\frac{p(Z_i,Z_j)}{p(Z_i)p(Z_j)})\right] = 2 \times I(Z_i; Z_j)
\end{aligned}
\tag{2}
$$

Where $I(Z_i; Z_j)$ is the mutual information between $Z_i$ and $Z_j$. The third equality is due to Proposition 3.1. Since non-zero mutual information implies positive correlation, we see that the amount of confounding between $Z_i$ and $Z_j$ is directly proportional to the correlation between $Z_i$ and $Z_j$. Hence, we use the correlation as a measure of confounding between generative factors in the causal processes of the form 1(a). $\square$

Using the connection between the confounding and correlation in causal graph 1(a), our objective is to generate counterfactual data such that the resultant dataset after augmentation looks similar to the data obtained from a causal process where there is no confounding between generative factors (i.e., no paths of the from $Z_i \leftarrow C_j \rightarrow Z_k; \forall i, j, k$). Equivalently, our counterfactual data generation algorithm removes the spurious correlations between generative factors by marginalizing the causal arrows $C_i \rightarrow Z_j$ for some $i, j$. To understand how counterfactual instances break the correlations, consider the following definition.

**Definition 3.4.** *(Counterfactual (Pearl, 2009)). Given an observed instance $X$ whose generative factors $Z_1, \ldots, Z_i, \ldots, Z_n$ take on the values $z_1, \ldots, z_i, \ldots, z_n$, the counterfactual instance $X'$ of $X$ (generated using the 3-step counterfactual inference procedure) differed from $X$ w.r.t. the generative factor $Z_i$, is an instance whose generative factors $Z_1, \ldots, Z_i, \ldots, Z_n$ take on the values $z_1, \ldots, z_i', \ldots, z_n$. Here $Z_i$'s value is changed from $z_i$ to $z_i'$ through an external intervention $do(Z_i = z_i')$.*

If we observe spurious correlation between two generative factors $Z_i, Z_j$ when they take on the values $z_i$ and $z_j$ respectively, generating counterfactual instances w.r.t. $Z_j$ with the intervention $do(Z_j = z_j')$ and adding the counterfactual instances to original data breaks the correlation between $Z_i, Z_j$. With this idea, we now present our algorithm to generate counterfactual images in a systematic manner remove confounding from observational data.

## 4 CONIC: METHODOLOGY

Our goal is to remove the effect of confounding in the observational data on a downstream task such as classification. To this end, we propose a way to systematically generate counterfactual data that can marginalize the effect of any confounding edge $C_i \rightarrow Z_j$ in Fig. 1 (a) as explained below.

**Removing The Confounding Effect of** $C_i \rightarrow Z_j$**:** In the causal graphs of the form 1(a), for paths of the form $Z_j \leftarrow C_i \rightarrow Z_l$, we call the edges $C_i \rightarrow Z_j$ and $C_i \rightarrow Z_l$ as confounding edges since together, their existence is the reason for confounding in the data. Also, let $(z_j^p, z_l^q)$ is one pair of attribute values taken by the variable pair $(Z_j, Z_l)$ under extreme confounding (e.g., in the training set of colored MNIST dataset, correlation coefficient of 0.99 between *color* and *digit* is observed such that whenever *color* is *red*, *digit* is 7 etc.). To remove the effect of the confounding edge $C_i \rightarrow Z_j$ w.r.t. the another confounding edge $C_i \rightarrow Z_l$ (recall that confounding between $Z_j, Z_l$ is present if and only if there exists a pair of causal arrows $C_i \rightarrow Z_j$ and $C_i \rightarrow Z_l$ for some $i$; due to this reason we consider the confounding effect of the confounding edge $C_i \rightarrow Z_j$ w.r.t. another confounding edge $C_i \rightarrow Z_l$), we consider two subsets $T_1, T_2$ of the observational data $\mathcal{D}$ which are constructed as follows. $T_1$ consists of the set of instances for which $Z_j \neq z_j^p$ and $Z_l = z_l^q$, $T_2$ consists of the set of instances for which $Z_j = z_j^p$ and $Z_l = z_l^q$. The size of $T_1$ is usually much smaller than the size of $T_2$ because of high correlation between $Z_j$ and $Z_l$ (e.g., there are more *red* 7's than *non-red* 7's).

Now, we learn a mapping $\mathcal{M}$ from the set $T_1$ to the set $T_2$ that changes the attribute $Z_j$ while fixing the value of $Z_l$ at $z_l^q$. That is, for any given instance $X \in T_1$, for which $Z_j \neq z_j^p$, $\mathcal{M}$ maps $X$ to a different instance $X'$ in which the value of the generative factor $Z_j$ is changed to $z_j^p$ (e.g., $\mathcal{M}$ takes *red* 9 as input and returns *red* 7 as output). This mapping $\mathcal{M}$ can be thought of as a function performing the 3-step counterfactual inference: learning the underlying generative factors, performing the intervention $do(Z_j = z_j^p)$ and then generating the counterfactual instance $X'$. Now, given an instance $X$ for which $Z_j \neq z_j^p$ and $Z_l \neq z_l^q$, using $\mathcal{M}$, we can generate counterfactual instance $X'$ in which $Z_j = z_j^p$ and $Z_l \neq z_l^q$. These counterfactual instances, when augmented with the original observed dataset $\mathcal{D}$, removes the effect of the confounding edge $C_i \rightarrow Z_j$ w.r.t. the edge $C_i \rightarrow Z_l$. That is, the counterfactual instances, when augmented with original data, breaks the correlation between $Z_j$ and $Z_l$. This process can now be repeated systematically for each confounding edge to generate counterfactual instances that remove the spurious correlations. Such augmented data points which differ from original data points w.r.t. only one feature (e.g., if original image is a *male* with *blond* hair color, augmented image is same *male* with *black* hair color) are referred as *coupled sets* by (Goel et al., 2021), *images generated by causal essential transformations* by (Wang et al., 2022). The overall procedure to generate counterfactual instances is summarized in Algorithm 1.

Earlier works use CycleGAN to generate counterfactual images that differ from original image by a single attribute/feature (Wang et al., 2022; Goel et al., 2021). Given two domains/sets of images that differ w.r.t. only one generative factor $Z_j$, a CycleGAN can learn to translate between the two domains by changing the attribute value of $Z_j$. In this case, one can think of CycleGAN as a function performing the required intervention $Z_j$ and generating counterfactual instance without modeling the underlying causal process. Concretely, CycleGAN is an architecture used to perform unsupervised domain translation using unpaired images. In a CycleGAN, a generator $G_1$ first transforms a given image $X$ from a

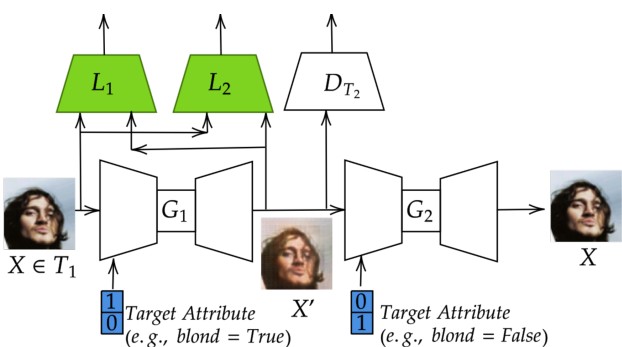

Figure 2: Architecture of the proposed modified conditional CycleGAN to generate counterfactual images. Pre-trained modules are shown in green color and target attribute is shown in blue color. Note that, for simplicity, we only show one pass of conditional CycleGAN (translation from $T_1$ to $T_2$) in this figure.

domain/set $T_1$ into $X'$ so that $X'$ appears to come from another domain/set $T_2$ such that certain features from input $X$ are preserved in the output $X'$. A discriminator $D_{T_2}$ then classifies whether the translated image $X'$ is original (i.e., sampled from $T_2$) or fake (i.e., generated by $G_1$). A second generator $G_2$ transforms the image $X'$ back to original image $X$ to ensure that $G_1$ is using the contents of $X$ to generate $X'$. The same procedure is repeated to translate images from domain $T_2$ into domain $T_1$. The loss function of CycleGAN can be written as follows.

$$\mathcal{L}_{CycleGAN} = \mathcal{L}_{GAN}(G_1, D_{T_2}, X, X') + \mathcal{L}_{GAN}(G_2, D_{T_1}, X', X) + \mathcal{L}_{cycle}(G_1, G_2) \quad (3)$$

---

**Algorithm 1:** Counterfactual Generation to Remove the Effect of Confounding Edge $C_i \rightarrow Z_j$

---

> **Result:** Counterfactual images that remove the confounding effect caused by the edge $C_i \rightarrow Z_j$
> **Input:** $\mathcal{D} = \{X_i\}_{i=1}^N$, Nodes $= \{Z_l | C_i \rightarrow Z_j \& C_i \rightarrow Z_l\}$
> **Initialize:** cf_images $= []$
> **for** each $Z_l \in$ Nodes **do**
>    $T_1 = \{X \in \mathcal{D} | Z_j \neq z_j^p \& Z_l = z_l^q\}$ \\* *divide data into two domains using attribute values* * \\
>    $T_2 = \{X \in \mathcal{D} | Z_j = z_j^p \& Z_l = z_l^q\}$
>    $\mathcal{M} =$ conditional CycleGAN$(T_1, T_2)$    \\* *Learn $\mathcal{M}$ to translate $T_1$ to $T_2$* * \\
>    Factual_Imgs $= \{X \in \mathcal{D} | Z_j \neq z_j^p \& Z_l \neq z_l^q\}$    \\* *Pick factual images from train set* * \\
>    CFs $= \mathcal{M}($Factual_Imgs$)$    \\* *Generate counterfactuals using $\mathcal{M}$* * \\
>    Append CFs to cf_images
> **end for**
> return cf_images

---

Where $\mathcal{L}_{GAN}$ is simple Generative Adversarial Network (GAN) (Goodfellow et al., 2014a) loss and $\mathcal{L}_{cycle}$ is cycle consistency loss measuring how well the output of $G_2$ is matching with the original input $X$. For example, $\mathcal{L}_{cycle}(G_1, G_2) = \mathbb{E}_{X \sim \mathcal{D}}[\|G_2(G_1(X)) - X\|_1]$ can ensure that $G_2(G_1(X)) = X$. In this work, to learn the mapping function $\mathcal{M}$, we use conditional variant of CycleGAN to leverage the supervision in terms of attribute values. For each generator, along with input, we also feed a desired target attribute as shown in the Figure 2.

To improve the quality of counterfactual images generated by conditional CycleGAN under extreme confounding, we propose a modification to conditional CycleGAN as detailed below. As discussed earlier, $X'$, the output of $G_1$, can be thought of as a counterfactual image of $X$. When changing the feature $Z_j$ of $X$, we keep the feature $Z_l$ fixed. That is, the representation for $Z_j$ in both $X$ and $X'$ should be different and the representation for $Z_l$ in both $X$ and $X'$ should be same. To ensure this, as shown in Figure 2, along with two generators $G_1, G_2$ and a discriminator $D_{T_2}$ that are part of conditional CycleGAN, we add two pre-trained discriminators $L_1, L_2$ (shown in green color in Fig. 2). $L_1$ takes two images $X, X'$ as input and returns high penalty if the representation of $Z_j$ is similar in $X, X'$ and small penalty otherwise. $L_2$ takes two images $X, X'$ as input and returns high penalty if the representation of $Z_l$ is different and small penalty otherwise. Thus, our overall objective to generate good quality counterfactual images is to train the modified conditional CycleGAN by minimizing the following objective.

$$\mathcal{L}_{conic} = \mathcal{L}_{CycleGAN} + \alpha(-\mathcal{L}_{contrastive}(L_1(X), L_1(G_1(X))) + \mathcal{L}_{contrastive}(L_2(X), L_2(G_1(X)))$$
$$- \mathcal{L}_{contrastive}(L_1(X'), L_1(G_2(X'))) + \mathcal{L}_{contrastive}(L_2(X'), L_2(G_2(X'))))$$
$$(4)$$

Where $\alpha$ is a hyperparameter and $\mathcal{L}_{contrastive}$ is the contrastive loss (Hadsell et al., 2006). For a pair of images $(X, X')$, $\mathcal{L}_{contrastive}$ defined as follows.

$$\mathcal{L}_{contrastive}(X, X') = AD^2 + (1 - A)\max(\epsilon - D, 0)^2 \qquad (5)$$

Where $A = 1$ if $X, X'$ belong to same class (or have same attribute values), $A = 0$ if $X, X'$ belong to different classes (or have different attribute values). $D$ is the distance between the representations of $X, X'$ (e.g., Euclidean distance). $\epsilon$ is the margin of error allowed between two representations of the images of different classes. $L_1$ and $L_2$ are pre-trained models and the parameters of $L_1$ and $L_2$ are fixed. That is, the loss values returned by $\mathcal{L}_{contrastive}$ are only used to update the trainable parameters of conditional CycleGAN.

**A Downstream Task - Image Classification:** To measure the goodness of counterfactual generation under confounding using Algorithm 1, we consider the classification task on the unconfounded test set as a downstream task. Let $\mathcal{D}^{aug} = \{(X_i, Y_i)\}_{i=1}^M$ be the dataset consisting of original data points from $\mathcal{D}$ and corresponding counterfactual data points. Usual empirical risk minimizer minimizes the following loss over $\mathcal{D}$.

$$\mathcal{L}_{erm} := \mathbb{E}_{(X,y) \sim \mathcal{D}}[l(f_\theta(X), y)] \qquad (6)$$

Where $l$ is cross entropy loss. Using $\mathcal{D}^{aug}$, we minimize the following loss $\mathcal{L}_{aug}$:

$$\mathcal{L}_{aug} := \mathbb{E}_{(X,y) \sim \mathcal{D}^{aug}}[l(f_\theta(X), y)] \qquad (7)$$

| Model | CM-MNIST | DCM-MNIST | WLM-MNIST | CelebA |
|---|---|---|---|---|
| ERM | 46.41± 0.81% | 43.31 ± 2.30% | 28.28 ± 0.70% | 70.64 ± 6.93% |
| CGAN | 41.86 ± 1.79% | 30.66 ± 3.86% | 17.50 ± 0.85% | 70.99 ± 2.35% |
| CVAE | 49.58 ± 1.50% | 41.99 ± 1.10% | 34.19 ± 1.58% | 71.50 ± 1.82% |
| C-$\beta$-VAE | 51.22 ± 1.00% | 51.58 ± 2.36% | 33.90 ± 1.87% | 74.29 ± 0.65% |
| AugMix | 47.36 ± 0.01% | 44.85 ± 0.02% | 26.30 ± 1.30% | 71.93 ± 4.64% |
| CutMix | 20.44 ± 1.22% | 23.10 ± 2.98% | 12.08 ± 1.59% | 73.66 ± 0.76% |
| IRM | 55.25 ± 0.89% | 49.71 ± 0.71% | 50.26 ± 0.48% | 72.30 ± 2.71% |
| CGN | 42.15 ± 3.89% | 47.50 ± 2.18% | 43.84 ± 0.25% | 69.25 ± 0.29% |
| CONIC | **65.57 ± 0.34%** | **92.41 ± 0.26%** | **77.72± 1.00%** | **79.56 ± 1.28%** |

Table 1: Test set accuracy results on MNIST variants and CelebA

To further improve the performance of a classifier using $\mathcal{D}^{aug}$, for each pair of images $X_i, X_j$ we minimize the contrastive loss $\mathcal{L}_{contrastive}(X_i, X_j)$ on the logits in the final layer. Now, the final objective to optimize for classification task is to minimize the following loss.

$$\mathcal{L} = \mathcal{L}_{aug} + \lambda \mathbb{E}_{(X_i, X_j) \sim (\mathcal{D}^{aug} \times \mathcal{D}^{aug})}[\mathcal{L}_{contrastive}(X_i, X_j)] \tag{8}$$

Where $\lambda > 0$ is a regularization hyperparameter.

## 5 EXPERIMENTS AND RESULTS

In this section, we present the experimental results on both synthetic (MNIST variants) and real world (CelebA) datasets. Having access to the ground truth generative factors (i.e., $Z_1, \ldots, Z_n$) of images,we artificially create confounding in the training data and we leave test data to be unconfounded (i.e., no correlation among generative factors). We compare CONIC with various baselines including traditional Empirical Risk Minimizer (ERM), Conditional GAN (CGAN) (Goodfellow et al., 2014a), Conditional VAE (CVAE) (Kingma & Welling, 2013), Conditional-$\beta$-VAE (C-$\beta$-VAE) (Higgins et al., 2017), AugMix (Hendrycks et al., 2020), CutMix (Yun et al., 2019), Invariant Risk Minimization (IRM) (Arjovsky et al., 2019), and Counterfactual Generative Networks (CGN) (Sauer & Geiger, 2021). To check the goodness of each of these methods, we check how well the performance of the downstream classifier on the test set is improved using the augmented images.

**MNIST Variants:** We construct the following three synthetic datasets based on MNIST dataset (Lecun et al., 1998) and its colored, texture, and morpho (where the digit thickness is controlled; Fig. 3) variants (Arjovsky et al., 2019; Castro et al., 2019; Sauer & Geiger, 2021): (i) colored morpho MNIST (CM-MNIST), (ii) double colored morpho MNIST (DCM-MNIST), and (iii) wildlife morpho MNIST (WLM-MNIST). We consider extreme confounding among generative factors as explained below.

For the experimental results shown in Table 1, in the training set of CM-MNIST dataset, the correlation coefficient between digit label and digit color $r(label, color)$ is 0.95 and the digits from 0 to 4 are thin and digits from 5 to 9 are thick (see Figure 3). That is, $r(label, thin) = 1$ if the digit is in [0,1,2,3,4] else $r(label, thick) = 1$. In the training set of DCM-MNIST dataset, digit label, digit color, and background color jointly take a fixed set of values 95% of the time. That is, $r(label, color) = $

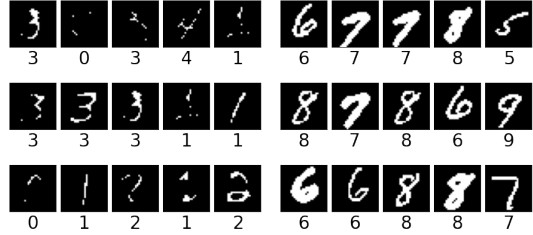

Figure 3: Left: sample thin morpho MNIST images and corresponding labels. Right: Sample thick morpho MNIST images and corresponding labels.

$r(color, background) = r(label, background) = 0.95$ and the digits from 0 to 4 are thin and digits from 5 to 9 are thick. In the training set of WLM-MNIST dataset digit shape, digit texture, and background texture jointly take a fixed set of attribute values 95% of the time and the digits from 0 to 4 are thin and digits from 5 to 9 are thick.

In all of these MNIST variants, test set images are unconfounded (e.g., in the test set of DCM-MNIST, any digit can be thin or think, can be in any background color, can be in any foreground color). In these experiments, under extreme confounding, our goal is to generate counterfactual images that break the confounding among generative factors. We evaluate models on this grounds by training a classifier using the augmented data and testing it on the unconfounded test data. Table 1 shows the results in which CONIC outperforms all the baselines. See Appendix for comparison of augmented images by various baselines. Coninc uses only 10000, 15000, 15000 counterfactual images in CM-MNIST, DCM-MNIST, and WLM-MNIST experiments as augmented images respectively to get improved performance. The regularization hyperparameter $\lambda$ in Equation 8 set to 0.5 for all MNIST experiments.

**CelebA:** Unlike MNIST variants, CelebA (Liu et al., 2015) dataset implicitly contains confounding (e.g., the percentage of *males* with *blond hair* is different from the percentage of *females* with *blond hair*, in addition to the difference in the total number of *males* and *females* in the dataset). In this experiment, we consider the performance of a classifier trained on the augmented data that predicts hair color given an image. Our test set is the set of *males* with *blond hair*.

We train models on the train set and test the performance on the set of *males* with *blond hair*. Since the number of *males* with blond hair is very low in the dataset (approximately 4% of *males* have *blond hair*), we show that the augmenting the train set with only 10000 images of *males* with *blond hair* improves the performance over baselines (see Table 1) whereas other baselines require more

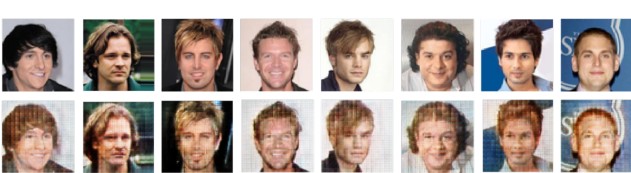

Figure 4: Top: CelebA original images of *males* with *non-blond hair* color. Bottom: Counterfactual images of *males* with *blond hair* generated using Algorithm 1

than 50000 augmented images to get minor improvement over ERM. Given a *male* image with *non-blond hair*, CONIC generates the counterfactual image with *blond hair* without changing the *male* attribute (see Figure 4 for sample counterfactual images). We also note that the deterministic models such as CGN fail when they are applied to a different task where the number and type of generative factors are not fixed and are difficult to separate (e.g., CelebA). CGN results in table 1 are obtained with only 1000 counterfactual images as augmented data points. When we increase the number of counterfactual instances, performance of CGN reduces further.

**Time Complexity Analysis:** Apart from its simple methodology, CONIC brings additional advantages in terms of computing time required to train the model that generates counterfactual images. As shown in Table 2, the time required to run our method to generate counterfactual images w.r.t. a generative factor $Z_j$ is significantly less than CGN that learns deterministic causal mechanisms as discussed in Section 2. Even though we used Cycle-GAN in this work, for the cases where the number of generative factors are more, StarGAN (Choi et al., 2018) can be used to minimize the time required to learn the mappings from one domain to another domain (Wang et al., 2022; Goel et al., 2021).

| Dataset | CONIC | CGN |
|---|---|---|
| CM-MNIST | $2.76 \pm 0.19$ | $103 \pm 1.50$ |
| DCM-MNIST | $2.22 \pm 0.01$ | $103 \pm 2.04$ |
| WLM-MNIST | $1.22 \pm 0.01$ | $111 \pm 2.50$ |

Table 2: Run time (in minutes) of CONIC compared to CGN on MNIST variants

## 6 CONCLUSIONS

We studied the adverse effects of confounding in observational data on the performance of a classifier. We showed the relationship between confounding and correlation in the causal processes considered, and we proposed a methodology to remove the correlation between the target variable and generative factors that works even when the dataset is highly confounded. Specifically, we proposed a counterfactual data augmentation method that systematically removes the confounding effect rather than addressing the confounding problem through random augmentations. Using the generated counterfactuals leads to substantial increase in a downstream classifier's accuracy. That said, we observed that the counterfactual quality can still be improved, which will be interesting future work.

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
