# OpenReview forum: "Counterfactual Generation Under Confounding"
_ICLR.cc/2023/Conference — Submitted to ICLR 2023_

### Official Review · Reviewer_8VNe · 2022-10-23

**Confidence:** 4
**Correctness:** 4
**Technical Novelty And Significance:** 2
**Empirical Novelty And Significance:** 2
**Recommendation:** 5

**Clarity, Quality, Novelty And Reproducibility:**

- Clarify: The paper was a bit hard to follow at times, especially starting from Section 4. It would be helpful to better separate out and organize the background information on CycleGAN, the methodology (the addition of the contrastive loss terms), the example with the two sets T_1, T_2, etc. I would recommend re-writing the paper to be more concise.
- Quality: The quality could definitely be improved, especially with respect to the empirical results.
- Novelty: The idea of using a CycleGAN for counterfactual generation in itself is not novel, though its use case for the presence of confounding is interesting.
- Reproducibility: See one of my points in the “Weaknesses” section. While there was a Jupyter notebook attached with the CelebA experiment, it did not include any additional information about the other baselines.


**Strength And Weaknesses:**

Strengths:
- Counterfactual generation is an important problem, and although there are works that try to address this in simple settings (e.g. no confounding), the setting in the presence of confounding is not as well-studied despite being more practically relevant and useful.
- I liked the earlier exposition about how the particular structure of the underlying causal mechanism (Figure 1a) directly allows for measuring confounding via mutual information estimation between the generative (latent) factors. It would have been really nice if the authors could have incorporated this into the experiments somehow (e.g. by looking at how the performance of CONIC is affected by the level of confounding as measured by this metric).

Weaknesses:
- I wasn’t able to find any further references back to directed information and how it could be used to measure the amount of confounding in the dataset.
- The empirical results left much to be desired, especially with the lack of detail regarding the experimental setups and ablations. For example, while CONIC seemed to improve the accuracy of the downstream classifier, I was surprised to see how poorly other approaches performed, such as the CGAN.
- I actually have a lot of clarification questions for the authors:
  - Was the discrepancy in performance a function of the architecture used (was it comparable in size to the CycleGAN)?
  - Were the conditional samples generated just purely by conditioning on each of the attributes in an attempt to “balance out” the training set?
  - Also, how much of the performance boost could be attributed to the additional contrastive loss in Eq. 8 of the downstream classifier rather than just standard ERM training using the real+counterfactual data as in Eq. 7?
  - In a similar vein, how much did each of the additional contrastive loss terms help in the CONIC training objective?
  - What do the samples from the vanilla conditional generators look like? Did those baselines also use any additional forms of supervision, such as a modified version of Eq. 8 (modified since they are not exactly counterfactuals)?
- ^ These details would be important to address in the final version of the paper, as it’s hard to tell what exactly the authors did and where the gains are coming from. This is the primary reason why it’s hard for me to recommend acceptance of the paper in its current form.
- It would have been nice to see whether the quality of the counterfactual images that were generated had any effect on the downstream results. For example, it is not obvious that there is a clear relationship between image quality and usefulness to the downstream task (https://arxiv.org/pdf/1905.10887.pdf), and the generated images in Figure 4 actually do not appear to be very high quality. Would the authors comment on this? I would have expected the downstream classifier trained on such images to perform worse on the test set, since the counterfactual images appear to be pretty out-of-distribution when compared to the images in the training set.
- The paper is also missing some relevant references (though these do counterfactual generation without confounding):
  - https://arxiv.org/pdf/2201.09119.pdf
  - https://arxiv.org/pdf/2202.10166.pdf

**Summary Of The Paper:**

This paper proposes a method called CONIC for generating counterfactual data under the presence of confounders, using a variant of CycleGAN where the training objective is augmented with a contrastive loss. The high-level idea is to have the CycleGAN generate a counterfactual X’ with respect to X such that it only modifies the attribute value of interest. The training data is then augmented with their corresponding counterfactual images, and this augmented dataset is used to train a downstream classifier which is evaluated on a test set without confounding. CONIC helps boost performance over existing baselines.

**Summary Of The Review:**

The paper proposes a method to generate counterfactuals under confounding using a modified version of CycleGAN, but does not provide enough empirical evidence to explain why/how the method is working. The baselines CONIC outperforms appear weak and are not well-explained in the paper, so it's hard to tell how strong they are. The clarity of the paper can also be improved. I am happy to raise my score if the authors can address these issues, though, since I think this work is interesting and addresses an important problem.

---

### Official Review · Reviewer_ZkeM · 2022-10-25

**Confidence:** 4
**Correctness:** 3
**Technical Novelty And Significance:** 3
**Empirical Novelty And Significance:** 3
**Recommendation:** 5

**Clarity, Quality, Novelty And Reproducibility:**

The writing of this submission is clear in general but there are a few issues which make a bit confusing as discussed above. The originality is solid. Regarding the reproducibility, I feel the algorithms details are not clear to me based on the descriptions in the submission.

**Strength And Weaknesses:**

## Weaknesses

1. How to get the two datasets for CycleGAN training? In my opinion, this requires knowing the attribute difference between the two datasets, which is related to the prior of data generating process (DGP). Does this mean one need to know about the DGP? If so, is it fair to compare it with algorithms such as IRM who does not assume such knowledge? If not, I think there are many larger scale datasets that should be taken into account (e.g. DomainBed). If I miss any point here, please feel free to correct me!

2. The datasets used in experiments are very spurious, in the sense that 95% of the data are spurious correlated. Therefore, I am curious to see how ERM perform if only trained with the rest 5% unbiased data, which I think is a more proper baseline if one knows which part of data is biased / spurious correlated.

3. What is the connection between the theoretical discussion in Sec 3 and the proposed method? I don't see any usage of mutual information term in the proposed algorithm, which is a bit confusing.

4. In Fig. 1(a), CONIC has multiple confounding factors $C_i$, which is considered as a major difference with previous work CGN. However, is this reflected in the design of the algorithm? Which part of the algorithm utilizes such an assumption?

### Minors
Some of the citations should use \citet{} instead of \citep{}.

**Summary Of The Paper:**

This paper introduces an algorithm to augment the classifier training with counterfactual examples. The proposed method modifies the CycleGAN algorithm to do this augmentation. Different from previous methods which could only deal with texture / shape, this algorithm could process any confounding effects to capture data-generation process.

**Summary Of The Review:**

The idea of this work makes sense in such a problem. However, this work suffers from some unspecified points discussed above. I would of course consider raising the score if they could be addressed properly.

---

### Official Review · Reviewer_YnUp · 2022-10-25

**Confidence:** 3
**Correctness:** 4
**Technical Novelty And Significance:** 4
**Empirical Novelty And Significance:** 2
**Recommendation:** 6

**Clarity, Quality, Novelty And Reproducibility:**

The paper overall is written well and is a useful read for understanding how where augmenting with counterfactual data helps. The way the writing is formatted makes it hard to read, with the many caveats in braces.

The simple modification seems like a valuable change to modeling variability in samples across groups.



**Strength And Weaknesses:**

Strengths:
1. The experiments show sizeable improvements to other counterfactual generation type methods.
2. The method looks complex but it is intuitively simple and I imagine the implementation to not be hard.

Weaknesses:

1. The paper does not discuss the value of counterfactual data for robust modeling, compared to other methods that do not build generative models using group-annotations in training. The authors should discuss this papers like https://arxiv.org/pdf/2110.14503.pdf and explain the value of their work within the larger ML goal of building robust models, especially without the additional and complex task of building a generative model.
2. The paper also does not discuss other work pointing out potential issues with counterfactually augmented data. Discussing those also would be worthwhile: https://arxiv.org/pdf/2107.00753.pdf

I would be happy to revisit my score if the authors can address these.

**Summary Of The Paper:**

The paper focuses on building robust models using counterfactual data. The main idea is to modify the cycle-GAN architecture to use group-annotations in the training data as additional supervision to learn to modify specific features. This method seems to yield useful improvements to the task of robust modeling when using generative models. The proposed method seems to be able to handle confounding better than other methods just condition on feature labels.

**Summary Of The Review:**

While I think the methodological contributions to generating counterfactual data are useful, I'm not convinced about the value of the method for the larger ML goal of building robust models.

---

### Official Review · Reviewer_jJ3z · 2022-10-25

**Confidence:** 3
**Correctness:** 3
**Technical Novelty And Significance:** 2
**Empirical Novelty And Significance:** 2
**Recommendation:** 5

**Clarity, Quality, Novelty And Reproducibility:**

The theoretical originality is interesting but with a very strong assumption, and the algorithm is built upon an existing approach.

The authors may consider some proofreading.

- Please add punctuation after all the mathematical equations.
- Please consider improving the readability of Algorithm 1.
- Fix typos, e.g., 'of the from' on the top of page 5.
- The notation $I(Z_i; Z_j)$ is used without formal definition on page 5.
- Add space before ',CutMix (Yun et al., 2019)'.

**Strength And Weaknesses:**

*Strength*

1. This paper considers an important confounding issue in image classification from observed data.

2. The authors provide sound theoretical results to study the relationship between correlation and confounding.

3. Numerical studies show their promising prospect on well-known benchmarks.

4. The paper is well-written in general and easy to follow.

*Weaknesses*

1. My main concern is that this paper primarily assumed that $Z_i$ are causally independent of each other, i.e., there is no causal link among different $Z$s, in the DAG illustrated in Figure 1a. This means the only correlation among $Z$s is spurious owing to their common confounders $C$. In this way, they are able to utilize their proposed CONIC to remove the Effect of confounding edge. Yet, the causal independence among $Z$s is a quite strong assumption and needs further justification. The authors may also comment on the worse case when this assumption is violated.

2. The implementation details are not clear. For example, I wonder how the authors weighted the five different loss functions in Equation 4. Or are they equally weighted or using the same strategy in Equation 8? More specifications are required.

3. The main methodology is built upon CycleGAN. The authors may benefit from decoupling their counterfactual data augmentation part from CycleGAN and show their method is generally applicable to different image classification methods. A very natural question is: how much has the proposed method improved CycleGAN? Why not include CycleGAN for comparison in the experiments of benchmark?

**Summary Of The Paper:**

This paper considers the important confounding issue in image classification problems from the observational data. The authors built the connection between the confounding and the spurious correlation, by assuming that the generating factors are causally independent of each other. This result allows them to add counterfactual data augmentation in CycleGAN to remove the correlation between the target variable and generative factors. Numerical studies show their promising prospect over existing methods.

**Summary Of The Review:**

Overall, this paper considers an interesting problem of confounding issues in image classification and is easy to follow. The theoretical results are sound but with a very strong assumption. The algorithm is built upon an existing approach (CycleGAN) while necessary comparison to show the uplifts is missing.

---

### Decision · Program_Chairs · 2023-01-20

**Decision:**

Reject

**Justification For Why Not Higher Score:**

The empirical studies have many moving parts and would need more thorough and systematic evaluation of large datasets to reliably demonstrate the performance of the proposal.

**Justification For Why Not Lower Score:**

N/A

**Metareview: Summary, Strengths And Weaknesses:**

The paper builds a connection between confounding and spurious correlation and proposes counterfactual data augmentation to remove the correlation. The key idea is to rely on a model where generating factors are causally independent, and use a variant of CycleGAN where the training objective is augmented with a contrastive loss.

strength:

The paper works on the important problem of counterfactual generation in the presence of confounding.

The paper discusses how causal structure could inform the strengths of confounding via mutual information.

weakness:

The empirical studies would need to be strengthened significantly. While the ablation study in the rebuttal suggests that the technique could be doing something interesting, it would require a more thorough evaluation and empirical study to cleanly disentangle the systematic effect of the additional contrastive loss across large datasets where there is confounding/spurious correlations. It is also not clear whether the quality of the counterfactual images is necessarily related to the performance on downstream tasks. Overall, the empirical studies in its current form are not ready for publication.


**Summary Of Ac-Reviewer Meeting:**

N/A